# Role of Recent PCR Tests for Infectious Ocular Diseases: From Laboratory-Based Studies to the Clinic

**DOI:** 10.3390/ijms24098146

**Published:** 2023-05-02

**Authors:** Sunao Sugita, Hiroshi Takase, Satoko Nakano

**Affiliations:** 1Department of Ophthalmology, Kobe City Eye Hospital, Kobe 650-0047, Japan; 2Department of Ophthalmology & Visual Science, Graduate School of Medical and Dental Sciences, Tokyo Medical and Dental University, Tokyo 113-8519, Japan; 3Department of Ophthalmology, Oita University, Oita 879-5593, Japan

**Keywords:** ophthalmology, PCR, microbiology, uveitis, ocular samples, infections

## Abstract

Infectious uveitis is a vision-threatening condition that requires prompt clinical diagnosis and proper treatment. However, rapid and proper diagnosis in infectious uveitis remains challenging. Several examination tests, including polymerase chain reaction (PCR) tests, are transitioning from laboratory-based basic research-level tests to bedside clinical tests, and recently tests have changed to where they can be performed right next to clinicians. In this review, we introduce an updated overview of recent studies that are representative of the current trends in clinical microbiological techniques including PCR tests for infectious uveitis.

## 1. Introduction

A PCR test is a method that uses special reagents (e.g., primers, probes) to amplify and detect the genes of foreign microorganisms such as viruses, including, for example, the novel coronavirus (COVID-19), which has been raging in recent years. The tissues are collected by swabbing the subject’s nose and pharynx. Evidence of the virus can be detected several days before the onset of symptoms, and the test is mainly used to check whether the virus is present in the body at the time of the test (i.e., the test shows whether the virus is currently active in the body). The sensitivity of the COVID-19 PCR test is reported to be about 70% [1], and if there is no virus in the collected specimen, the test may be negative even if the patient is infected. For this reason, depending on the testing institution, cases in which the virus cannot be detected may be expressed as “not detected” rather than as “negative.” Generally, in PCR tests, it is necessary to be careful of false negatives. A false negative means that the PCR test is negative, but the patient actually has the infection. In clinical practice of ophthalmology, this PCR test is used for several reasons (including a small local ocular sample, among other reasons), and its sensitivity and specificity are high [2,3]. One of the advantages of the PCR test is that it is rapid, and the latest ophthalmological test can be completed in about 60 min [4]. In fact, there is a long history of PCR testing uncovering the etiology of a variety of ocular diseases of unknown etiology, i.e., ocular inflammation caused by infectious agents. A recent publication, “Adenovirus-associated Uveitis with Necrotizing Retinitis,” has greatly surprised ophthalmologists and may represent a breakthrough of a novel intraocular infection [5]. Similarly, in the field of ophthalmology, there have been two surprising discoveries in the past. The first was that HSV-2 is one of the pathogens that cause retinitis [6]. Herpes simplex virus (HSV) includes type 1 (HSV-1) and type 2 (HSV-2), and HSV-2 was thought to cause sexually transmitted diseases, but it was discovered that HSV-1 as well as HSV-2 cause necrotizing retinitis. The second was that cytomegalovirus (CMV) is one of the pathogens that cause corneal endotheliitis [7]. Both discoveries occurred before the development of the recent PCR examination (e.g., multiplex PCR). Due to these discoveries that have proven the usefulness of PCR, new methods and devices continue to be developed.

In this review, the characteristics of local eye specimens, including the quality of specimens for testing; methods for collecting local eye specimens; an introduction to PCR in the ophthalmic fields; interpretation of the results of local eye specimen testing; and the latest molecular biological methods for identifying infectious diseases are summarized. In particular, the multiplex PCR test for ophthalmology that we developed and its clinical application for ocular infectious disorders are introduced.

## 2. Characteristics of Ocular Samples, Quality of Samples, and Method of Sample Collection

### 2.1. Characteristics of Ocular Samples: Specimen Quality

The flow of a recent PCR test for ocular samples is shown in Figure 1. A recent trend in the field of ophthalmology is to perform multiplex PCR with graphical assessment (Figure 1). No matter how much technology advances, there are some things that cannot be neglected, specifically the “specimen quality.” In PCR testing of an infectious disease, the success or failure of the test can depend on whether a pathogen-rich specimen can be collected. The collection site/location, collection time, collection technique, and transportation/storage method are important. For example, in the case of a corneal ulcer in corneal infectious diseases, it is necessary to scrape not the center of the ulcer but between the normal part and the lesion that is expanding. Alternatively, if the lesion is extensive, the lesion itself should be scraped. The non-diluted vitreous, which contains as few impurities as possible, such as bleeding, lens, and perfusate, is preferable for examination, as well as the aqueous humor. Moreover, the timing of sampling is also important, and samples should be taken during the active phase of infection, i.e., when inflammation is actually present. In addition, in order to test from limited amounts of specimen, the clinical diagnosis is clarified before testing, and after the target pathogen and the differential pathogen are identified, the specimens are sorted into the necessary tests to be diagnosed, transported, and stored. It is necessary to proceed to sample collection only after proper planning. It should be kept in mind that pre-treatment specimens (e.g., prior to treatment with antibiotics/steroids) may affect PCR results.

### 2.2. Method of Sample Collection

Aqueous humor or vitreous fluid is used for PCR testing to investigate the causative pathogens of infectious uveitis. Aqueous humor is usually collected in amounts of around 100 to 200 µL by anterior chamber tap, either in the outpatient clinic or during intraocular surgery. In the outpatient setting, the ocular surface is anesthetized with oxybuprocaine hydrochloride and sterilized with 0.5% povidone-iodine. A 30-gauge needle with a tuberculin syringe or a disposable pipette with a 30-gauge needle [8] is used to tap through the corneal limbus (Appendix A). Vitreous fluid is usually collected during vitreous surgery. While the diluted vitreous in the drain bag of the vitrectomy machine can be applied for PCR, the undiluted vitreous is necessary for quantitative or semi-quantitative PCR. The undiluted vitreous can be collected with dry vitrectomy and scleral indentation before the intraocular irrigation is started. The vitreous fluid is cut and aspirated into the dry extension tube connected with a 10-cc syringe through a T-shape stopcock. After the intraocular irrigation has started, about 1.5 to 2 mL of the undiluted vitreous sample in the extension tube can be refluxed into a sample collecting tube or a syringe through the vitreous cutter by pushing the 10-cc syringe (Appendix A).

### 2.3. Interpretation and Precautions of PCR Results of Eye Specimen Tests

In many examination tests, the detected pathogen is not always the causing pathogen. It is necessary to “interpret” the obtained results with knowledge and “connect to the diagnosis” by considering various information such as clinical findings. This is the last remaining job for expert doctors, even if AI and machine learning develop. In particular, metagenomic analysis, which detects a large number of pathogens, requires the ability to organize and interpret obtained microbial information. Regarding PCR or culture, information on the number of pathogens (e.g., high copy numbers of a certain virus genomic DNA) provide important information in reaching the correct diagnosis. Development of multiplex PCR using local ocular specimens has made it possible to diagnose many ocular infections. In addition, herpes virus-related ocular infection can be detected at a rate of almost 100%, that is, definitive diagnosis can be made by PCR tests if appropriate sample collection is performed. “Appropriate specimen collection” is important. For example, in herpes virus corneal epitheliitis, the herpes virus is present in the corneal epithelium. Therefore, PCR testing of the aqueous humor would be unrevealing, due to cellular infiltration. Cellular infiltration within the anterior chamber is reactive inflammation, and the pathogen (herpes virus) is not present within the anterior chamber. In addition, there are infectious ocular diseases such as tuberculosis that cannot be diagnosed by PCR alone. Typical examples are *Mycobacterium tuberculosis* (tuberculous uveitis), *Treponema pallidum* (syphilitic uveitis), bartonella (cat-scratch disease), and toxocara (ocular toxocariasis). Intraocular tuberculosis in Japan is mainly an allergic reaction to the tuberculosis antigens, and PCR examination is not useful [9]. PCR may be positive in some cases of syphilis, but caution is required because the aqueous humor tends to be negative.

As for Bartonella, even in cases of typical cat-scratch disease (CSD), the authors have no experience with PCR-positive intraocular fluids and submit to the Bartonella serum antibody test. It is assumed that the PCR test is not useful in CSD patients since the pathogen might be hidden deep in the retinal tissue/vessels. As with tuberculosis, toxocariasis is also considered to be primarily an allergic reaction, and steroid treatment is effective. The parasite-related uveitis caused by *Toxoplasma gondii* has DNA that can be easily detected in the aqueous humor and vitreous [2,3], and their pathologies are very different. On the other hand, antibody testing is more effective than PCR for *Toxocara*. Therefore, it is necessary to decide whether to conduct the test while knowing that there are cases of PCR false-negatives.

Another issue with PCR testing is that there are often false-positive cases. Identification of bacterial and fungal species by PCR is always accompanied by the problem of contamination, including contamination during specimen collection and contamination during specimen processing (due to contamination of the technical staff or testing space). In addition, reagents may contain traces of bacterial DNA. Maximum caution is required when testing for bacterial and fungal species. Bacterial 16S ribosomal RNA (rRNA), fungus 28S/18S rRNA gene, and *Cutibacterium acnes* (*C. acnes*) demonstrated that setting cut-off values (positive, suspected positive, and negative values) was necessary due to the problem of contamination [10]. Validation of PCR machines in each institute and education of the staff are mandatory. For example, devising the flow line in the inspection room, applying UV light to the work room and cabinets, and frequently changing gloves are all practical measures to prevent contamination. It should also be remembered that the gold standard for testing for these bacterial and fungal species is culture and smear testing, and PCR is only a complementary diagnostic tool. Moreover, although the inside of the eye is sterile, the ocular surface is not sterile and contains commensal bacteria. In infectious diseases of the ocular surface, it is important to distinguish whether the bacterium is the causative bacterium of the inflammation or whether the bacterium is just a commensal organism.

In addition to issues of false-positive or false-negative results by PCR, we must think about the line between positive and negative results. If ocular tissues and infiltrating cells are latently infected with a virus (e.g., herpesviruses), they are detected by PCR even though they are not pathogens. When quantitative PCR is performed, it is “positive” when high-copy-number DNA is present in the sample. However, in the case of low copy numbers, we are unable to distinguish between positive and negative results in order to diagnose the infectious pathogens. However, this is not only a problem with PCR tests but with other tests as well (e.g., virus antigen tests). Rather than using the results of PCR alone for diagnosis, we believe that characteristic ocular findings with active inflammation, other test results, and other information such as treatment effects should be comprehensively evaluated.

## 3. Examination of Multiplex and Broad-Range PCR in Ophthalmology

PCR is used as a test to diagnose or rule out infectious eye diseases such as uveitis. A current clinically viable test method for PCR testing in ophthalmology, specifically multiplex and broad-range PCR, as well as a novel PCR termed “Strip PCR” for novel multiplex PCR, have recently been developed. The greatest feature of multiplex PCR is its ability to simultaneously and rapidly detect several numbers of infectious antigen genomes.

The initial PCR kit was made to detect 24 pathogens. Its infectious antigens include viruses, bacteria, fungi, and parasites. Specifically, it detects pathogens that cause infectious ocular diseases, such as following: herpes simplex virus (HSV) type 1, HSV2, varicella-zoster virus (VZV), Epstein-Barr virus (EBV), cytomegalovirus (CMV), human herpes virus (HHV) 6, HHV7, HHV8, human T-cell lymphotropic virus (HTLV)-1, human adenovirus (HAdV), *M. tuberculosis*, *Treponema pallidum*, *C. acnes*, bacterial 16S rRNA, *Candida* species, *C. glabrata*, *C. krusei*, *Aspergillus*, *Fusarium*, fungal 28S rRNA, *Toxoplasma*, *Toxocara*, *Chlamydia trachomatis*, and *Acanthamoeba* [11,12,13]. The cases of patients with infectious ocular diseases of unknown cause are tested using this PCR kit.

Broad-range PCR amplifies and diagnoses common gene regions of bacterial and fungal species: several microbiological techniques exist for bacterial pathogen identification in endophthalmitis using 16S rRNA pan-bacterial PCR [11,12,13]. PCR was designed with bacterial 16S rRNA-specific primers and probes to detect the bacterial 16S rRNA gene region [3]. For PCR amplification of the 16S rRNA gene, about 500 base pairs (bps) of the first half of the 16S rRNA are analyzed. During specimen collection and the PCR procedure, careful attention to bacteria and bacterial DNA contamination is needed, particularly laboratory contamination. In order to detect bacteria species, we also perform Basic Local Alignment Search Tool (BLAST) analysis in some of the cases. BLAST refers to an algorithm for performing sequence alignment of DNA base sequences in bioinformatics and refers to a program implementing the algorithm. By using BLAST to search one’s own sequence against the sequence database, groups of similar sequences with scores above a certain number can be located. The amplified PCR product is directly sequenced using the ABI analyzer and evaluated for homology with reference sequences using NCBI BLAST. Bacterial sequences with 99–100% homology are identified (98% is also acceptable). In addition, we conducted broad-range real-time PCR analyses of the 28S rRNA gene or 18S rRNA gene region from fungi species such as Candida and Aspergillus, which are considered to be highly ophthalmologically relevant among fungal infections [3]. A feature of our broad-range PCR is that fluorescent probes can be mixed with reagents for quantitative evaluation. By quantifying the amount of genome (which is the copy number) in the sample, it is possible to determine the therapeutic effect of antibiotics and antifungal drugs.

One of the advantages is that this PCR is often positive even if the culture or smear is negative in cases where antibiotics have been administered before the test. Many ophthalmologists begin administering broad-spectrum antibiotic eye drops as soon as they see a suspected infection (without waiting for test results). The patient in Figure 2 is a case of keratitis with frequent administration of antibiotic eye drops and systemic administration of antibiotics. Infectious keratitis was suspected based on the findings, but the culture and smear of the corneal specimen were negative, making it difficult to treat. A second corneal scraping was submitted for PCR testing and was positive for bacterial and fungal genomes. PCR can pick up dead bacterial DNA, so it may be possible to distinguish (and thus diagnose) even if a large amount of antibiotics has been administered in advance like this. In fact, the keratitis in this case was completely cured by changing antibiotics and using anti-fungal agents. The disadvantage is that unlike with culture, it is not possible to determine the type of bacteria species. Development of multiplex PCR specialized for infectious endophthalmitis and keratitis is awaited, for example, for detection of various bacteria such as *Staphylococcus aureus*, *Pseudomonas aeruginosa*, *Escherichia coli*, *Klebsiella*, *Enterococcus*, *Streptococcus pneumoniae*, and antibiotic resistance genes. If the sample is Staphylococcus aureus positive and mecA gene positive at the same time, “methicillin-resistant Staphylococcus aureus (MRSA)” can be diagnosed. We are currently undertaking development of a multiplex PCR kit for this bacterium.

In recent years, Strip PCR to detect 24 pathogens that cause ocular infections has been developed [10], and a Direct Strip PCR that skips DNA extraction specifically for antigens that cause infectious uveitis in Japan has also been developed [3,4]. This direct strip PCR test can test 9 items (HSV1, HSV2, VZV, EBV, CMV, HHV6, HTLV-1, *Treponema pallidum*, and *Toxoplasma*) from 24 pathogens. These nine can comprehensively test representative infectious diseases of infectious uveitis in Japan. This eliminates the step of extracting DNA from the sample, making it very convenient for the PCR operator and shortening the overall time. This PCR facilitates the identification of the etiology and is even more rapid, within 60 min, and is also used as a PCR diagnosis tool during vitrectomy at our hospital. The concept of PCR testing has been changed so that PCR results can be obtained during vitrectomy (for example, similar to intraoperative rapid pathological examination of cancer patients). Since ophthalmic local specimens are always very small, they are useful in this field where the amounts of specimens are always small. The Direct Strip PCR test developed in Japan specifically covers infectious antigens for Japan, but the test items can be changed according to the needs of other countries. For example, HTLV-1-associated uveitis is common in Japan, but it is not necessary to test for it in Europe and the United States because it is rare. On the other hand, tuberculosis, which is common in India and South America, should be added to the testing lineup. The Direct Strip PCR kit for infectious uveitis consists of only eight Strip PCR tubes and one DNA amplification buffer tube. Six PCR tubes on the eight-tube strip were precoated with enzymes, primers, and probes (6FAM and ROX fluorescence) targeting one to two different pathogens [4]. Because of an enzyme (for pre-treatment of the sample) in the kit, we can skip the DNA extraction in the sample. We are now conducting research on detecting bacteria and fungi by Direct Strip PCR in addition to the viruses. The validation is almost completed.

PCR, in which primers and probes are immobilized on tubes, is also being developed for use in regenerative medicine. Recent reports indicate a trend toward comprehensive PCR testing to identify mycoplasma infection in transplanted grafts [14]. In the field of regenerative medicine, there is a need for kits that can simultaneously and rapidly detect multiple infectious antigen genomes (e.g., viruses, bacteria, and fungi) in addition to mycoplasma infections. Implanting a graft exposed to infectious microorganisms would be a big issue for the patient, so we think that rapid and comprehensive PCR should be introduced into the screening test.

## 4. New Findings by Multiplex PCR in the Field of Ophthalmology

A wide variety of pathogens such as viruses are involved in ocular infections, and recent advances in molecular biology testing techniques have enabled diagnosis in many infectious cases. Multiplex PCR is useful for the diagnosis of herpes eye disease. Because of the PCR data and specific ocular findings, HSV1-, HSV2-, VZV-, and CMV-associated uveitis are able to be diagnosed. However, the involvement of EBV, HHV6, HHV7, and HHV8 in ocular disease was still unknown for a long time. Clinical application of multiplex PCR has contributed not only to uveitis but also to detection of infection after corneal transplantation [15]. In fact, there have been a few cases in which CMV infection was involved that were thought to be cases of rejection after corneal transplantation. Recently, patients with CMV-related corneal endotheliitis were reported after descemet membrane endothelial keratoplasty, and the findings closely mimic graft rejection [16]. After corneal transplantation, PCR testing of aqueous humor is advised for suspect cases (graft rejection versus CMV corneal endotheliitis). Although there have been many reports of EBV and uveitis, the pathogenesis of intraocular inflammation remains unclear [3,12]. On the other hand, reports of HHV6 ocular inflammatory patients are relatively recent. In our experience [17], 7 cases (2%) of 350 intraocular specimens from patients with ocular inflammation were positive for HHV6-DNA. In addition, 1/65 (1.5%) corneal tissue samples were positive for HHV6-DNA [17]. A recent report by another group found that a case of HHV-6-related corneal endotheliitis developed after intravitreal ranibizumab injections [18]. HHV6 mRNA was detected in intraocular samples from a patient positive for HHV6 genomic DNA positive and with active intraocular inflammation, suggesting the possibility of viral replication and reactivation in the eye [17]. Most of the HHV6-DNA in the intraocular fluid of inflamed eyes is thought to be the result of release from intraocular resident cells and infiltrated immune cells due to intraocular inflammation. These viruses infect the intraocular tissues, especially the retinal pigment epithelial cells, and then become dormant. Our consideration is that this viral genome detection is a “secondary factor”.

Ocular manifestations associated with HHV7 [19] and HHV8 [20] have been reported as corneal endotheliitis. These endotheliitis cases must be differentiated from those due to CMV infection. In addition, in cases that occur after corneal transplantation, it is necessary to differentiate the involvement of specific viruses in graft rejection.

The most recent novelty that multiplex PCR has yielded is adenovirus-associated uveitis/retinal necrosis [5]. Uveitis with retinal necrosis is a disease that causes vision loss and blindness and results in irreversible “Quality of Vision” deterioration in patients. Several pathogens and diseases are known to cause uveitis, but most have been thought to be caused by the herpesvirus. We recently reported the first two cases of uveitis with retinal necrosis caused by human adenovirus infection. As a result, adenovirus was detected by a multiplex PCR test using intraocular fluids from the uveitis with necrotic retinitis. Genome analysis of the adenovirus detected a new strain C type 6, which is rarely reported clinically, from case 1, and a new strain D type from case 2 [5]. Until now, about 35–40% of all uveitis cases were of unknown cause [21], but there is a possibility that some of them were caused by adenovirus infection. Although the route of adenovirus infection and the mechanism of retinal inflammation in these two cases remain unknown, this discovery is expected to contribute to the selection of appropriate treatment and the understanding of uveitis associated with retinal necrosis.

## 5. Recent Molecular Biological Methods for Identifying Infectious Diseases

Point-of-care testing (POCT) helps improve treatment and prognosis based on etiologic diagnosis through reduced testing time and on-site testing [22]. Here, we introduce representative recent molecular biological techniques for identifying infectious diseases (Table 1). They include immunochromatography, quantitative PCR, DNA microarray-based assay, multiplex PCR-based examination, and metagenomic next-generation sequencing (Table 1). We also explain whether these tests can be POCT in the clinic.

### 5.1. Immunochromatography

In the immunochromatography method, antigens in a sample form an immune complex with a labeled antibody and move across a cellulose membrane, where they react with a captured antibody and are determined by color. Immunochromatography is inexpensive, has short assay times (<15 min), and requires no special equipment or skilled technicians [23]. Therefore, immunochromatography is widely used as a POCT test for infectious diseases requiring rapid diagnosis. However, it has the disadvantages of being less sensitive than PCR and limited to handling one to two pathogens. The kit should also have a large number of pathogens in the sample—over 10^3^–10^5^ pfu/mL (Table 1).

### 5.2. Quantitative PCR

Quantitative PCR (qPCR) is a sequence-specific PCR method and is the most common procedure for direct identification of pathogens (Table 1). Because it is quantitative, it is useful for diagnosing pathological conditions and determining therapeutic efficacy. It usually requires nucleic acid extraction before PCR. The disadvantages of this method are that it is not comprehensive, it requires prior knowledge of the pathogen sequence, and it requires the design of assays specific to individual pathogens. It also usually requires time, labor, and skilled technicians.

### 5.3. Verigene^®^ and FilmArray^®^

DNA microarray-based Verigene^®^ (Luminex Corporation, Austin, TX, USA) and multiplex PCR-based FilmArray^®^ (bioMérieux, Marcy l’Étoile, France) simplified a complicated molecular biological method and are approved by the U.S. Food and Drug Administration (FDA); their commercial versions have been widely available for diagnosis in clinical microbiology laboratories for several years. They are simple and rapid with few manual operations and do not require a skilled technician. Pooled sensitivity and specificity estimates for detection of AMR determinants by the Verigene and/or FilmArray systems were 85.3% and 99.1%, respectively, across 15 studies, and 95.5% and 99.7%, respectively, across five studies [24,25]. However, these do not address ocular infectious pathogens, although there are kits for sepsis, respiratory infections, meningitis, and drug resistance genes. They also are not able to treat intraocular fluids, as they require large sample volumes (350 µL to 700 µL) (Table 1).

### 5.4. nCounter^®^ Analysis System

NanoString’s nCounter Analysis technology is a high-throughput and multiplexed direct digital detection system counting individual RNA and DNA molecules in a sample. A custom-designed comprehensive ocular panel for the nCounter^®^ Analysis System contained 46 pathogens and 2 resistance/virulence markers that are commonly detected in intraocular infections (Table 1). It had PCR-level sensitivity, and quantification of DNA hybrids by NanoString had good correlations with the real-time PCR Ct values. Because it required nucleic acid extraction and the total time was approximately 12 h, it is not suitable for POCT [26].

### 5.5. Direct Strip PCR^®^

Therefore, we have developed a new kit that can handle 20 µL trace intraocular fluids such as aqueous humor and vitreous fluids (Table 1). Direct Strip PCR^®^ consists of solid-phased multiplex real-time PCR assays without DNA extraction and measuring operation for trace reagents [9]. It is widely used in routine diagnostics of infectious uveitis, endophthalmitis, and kerato-conjunctivitis because of its high sensitivity, high specificity, and rapidity (<60 min). In a previous study (from 2015 to 2019, 18 domestic and international centers, 511 cases), the PCR exhibited higher repeatability and specificity, long storage stability, and detection ability equal to that of qPCR [4]. It also showed low interinstitutional variability compared with qPCR, even when PCR beginners used various real-time PCR machines. The Direct Strip PCR for nine uveitis pathogens exhibited high concordance with qPCR. In addition, results obtained using Direct Strip PCR and qPCR were highly correlated (ρ = 0.748; *p* < 0.001). With the development of a number of small PCR instruments that are compatible with COVID-19, this multiplex assay can now be used for POCT testing, including intraoperative rapid diagnosis (during vitrectomy). Based on a multicenter study of more than 1300 specimens, Direct Strip PCR included nine major pathogens suitable for PCR testing for infectious uveitis (HSV1, HSV2, VZV, HTLV-1, HHV6, EBV, CMV, *Toxoplasma gondii*, and *Treponema pallidum*), which were selected from 24 major ocular infectious disease pathogens [10]. The current disadvantage is that these pathogens were created for uveitis in Japan, and some improvements are necessary for use in other countries [3].

### 5.6. Metagenomic High-Throughput Next-Generation Sequencing (mNGS)

Only pre-identified pathogens can be detected by culture, immunochromatography, and PCR, including multiplex PCR, conventional Sanger sequencing as a low-throughput method based on dideoxynucleotide chain termination. However, mNGS that is independent of pathogen sequence will enable accurate detection of all pathogens in ophthalmic samples. mNGS may revolutionize the diagnosis of ophthalmic infectious diseases. NGS using Illumina short-read (~500 base pairs) sequencing platforms and NGS using nanopore very long-read (~1500 to 882 k bases) sequencing provide comparable data. The rapid diagnostics by portable, pocket-sized, relatively low-cost nanopore sequencers (Oxford Nanopore Technologies’ MinION sequencer) has been more suitable for cost-effective point-of-care diagnostics and real-time genomic surveillance [27]. Rubella virus infection of the eye is a known cause of Fuchs’ heterochromic iridocyclitis (FHI); NGS demonstrated the presence of rubella virus RNA in the eyes of FHI patients, while reverse transcription PCR was positive in the sample with the highest number of sequencing reads [28]. NGS not only identifies the initiating organisms but also the ocular surface microbiome that may be affected by disease, like the relationship between intestinal microbiota and bacteriophage [29]. The high-throughput capabilities of NGS result in an explosion of sequence data that must be interpreted. Metagenomic analysis requires proper interpretation of the data collected, and many issues remain, including diagnostic thresholds and contamination. It was necessary to remove known background contaminants from the data of potential pathogens for each subject [30]. The big issues are the high cost and costly maintenance. Because of this, POCT is not suitable; only lab-based assay is now available (Table 1).

## 6. Conclusions and Future Directions

PCR testing for ocular infections is gaining importance not only for diagnosing the etiology but also for excluding infectious diseases before administering steroids, immunosuppressants, and biological agents and before surgery. In addition, the latest simplified genetic testing for infectious diseases is useful not only at core hospitals such as university hospitals but also at clinics, which are on the front line of medical care. Moreover, multiplex PCR tests for unexplained uveitis and eye infections have revealed that pathogens such as viruses, which had not been known or thought to be involved, are involved in intraocular inflammation. We guess that in the near future, POCT will be a system that allows for simple and rapid testing on the spot where the specimen is collected, instead of moving the patient or specimen to the examination department. For example, in ophthalmology clinics, we are now in an era where aqueous humor is collected from uveitis patients who have given informed consent, multiplex PCR is performed on the spot, and the results can be obtained while the patient is waiting in the clinic.

In addition, for the advancement of infectious disease science in the field of ophthalmology, Koch’s postulates should not be forgotten in infectious disease research. In the pursuit of speed and simplicity, PCR-based pathogen detection methods have advanced. Isolation and culture efforts should not be spared in order to include various pathogens in textbooks as etiologic agents of specific diseases in the future. If possible, animal experiments are also necessary. Basic research on infectious diseases is extremely important for further development of clinical testing.

## Figures and Tables

**Figure 1 ijms-24-08146-f001:**
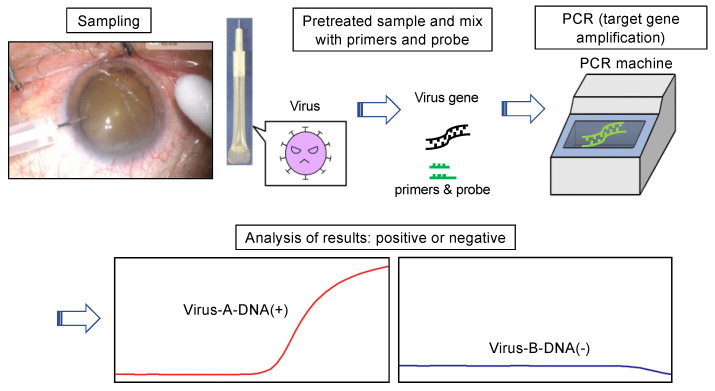
Flow of a PCR test for ocular samples. A PCR test is a test that shows whether the genome of an infectious antigen such as a virus is currently active in the body. It amplifies and detects the gene of the target virus using a PCR reagent. In the case of infectious uveitis, ocular sampling of subjects is performed. The upper left photo shows the collection of aqueous humor. After pre-treating the sample, it is then mixed with dedicated reagents (primers and probes) and applied to the PCR machine. Generally, the target viral gene is detected in a few hours, but the latest PCR kits provide results in <60 min. Results are determined by curve analysis such as graphs. The graph rises for positives (red) and remains flat for negatives (blue).

**Figure 2 ijms-24-08146-f002:**
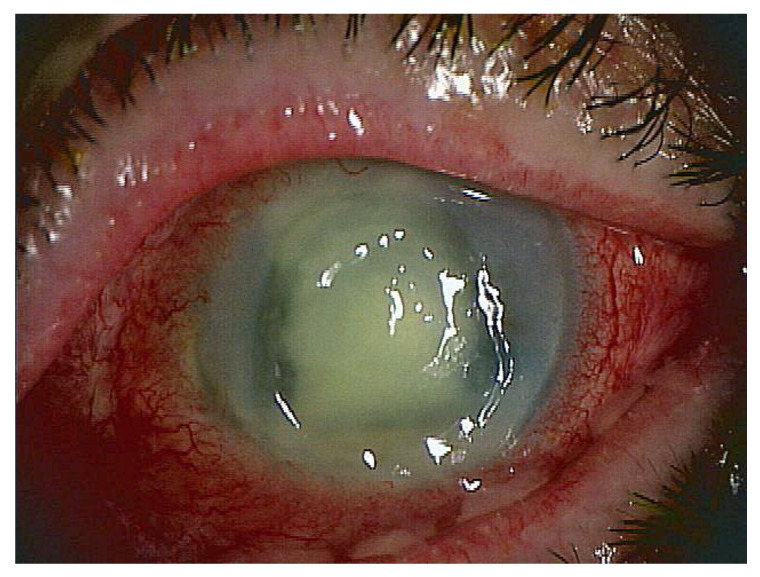
Slit lamp pictures of a case of keratitis (72 years old, male): Infectious keratitis was suspected. The panel is the corneal epithelial defect and opacity due to inflammation. PCR testing indicated co-infection with bacterial and fungal species. Eventually, the keratitis in this case was completely cured by changing antibiotics and using anti-fungal agents.

**Table 1 ijms-24-08146-t001:** Comparing the molecular biological methods for identifying infectious diseases.

	Immune-Chromatography	qPCR	Verigene^®^ (Microarray)	FilmArray^®^	nCounter^®^ Analysis System	Direct Strip PCR^®^	mNGS
Sample volume	100 µL	100 µL	350–700 µL	300 µL	100 μL	20 µL	50 µL
Total time	10 min	150 min	150 min	60 min	12 h	60 min	4 h
Pathogens	Limited	Limited	Limited	Limited	Limited	Limited	Comprehensively
Sensitivity	Low	High	High	High	High	High	Highest
Quantitative	No	Yes	No	No	Yes	Yes	Yes
Costs	Low	Middle	Middle	Middle	High	Middle	High
Equipment	Nothing	Small, inexpensive	Middle, expensive	Middle, expensive	Expensive	Small, inexpensive	Expensive
POCT	Suitable	No	Suitable	Suitable	No	Suitable	No
Reference			Need culture 24 h	Skip DNA extraction		Skip DNA extraction	

mNGS—Metagenomic next-generation sequencing; POCT—Point-of-Care Testing; qPCR—Quantitative PCR.

## Data Availability

Not applicable.

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
