# Peer review of "Role of Recent PCR Tests for Infectious Ocular Diseases: From Laboratory-Based Studies to the Clinic"

_ijms, 2023, doi:10.3390/ijms24098146_

Round 1
Reviewer 1 Report
The authors have made a comprehensive review on recent PCR progresses, especially with the detection of ocular infectious diseases, based on their studies and references review. They also pointed out the possible point-of care molecular diagnosis development in the near future. This review covers the main molecular diagnosis tests including Strip PCR, mNGS and many other modalities. Overall, this is a good review and should provide a comprehensive knowledge to the readers. I have some points to address:
Major:
1. Line 119: "As for Bartonella, even in cases of typical cat-scratch disease, the authors have no experience of PCR-positive intraocular fluids, and we are entrusted with the Bartonella serum antibody test." Beside their clinical experience, I think the authors should address a little more regarding previous evidence of the pathophysiology of CSD. Why is PCR mostly negative from vitreous/aqueous sampling ? (such as autoimmune reaction; molecular mimicry; or the pathogen is hidden deep in the retinal tissue ?)
2. Line 198: "Direct Strip PCR that skips DNA extraction" I suggest the authors cover a little more details on why DNA extraction can be skipped using this technique. Meanwhile, the reasons, pros and cons of not detecting bacteria and fungi by direct strip PCR should also be mentioned.
Minor:
1. Abstract: "Infectious uveitis is a vision-threatening emergency..." As some infections are slowly progressive and not necessarily fulminant, I suggest using a more neutral description: "vision-threatening condition"
2. Line 60: "to perform multiplex PCR and graphically assess" I suggest "to perform multiplex PCR with graphical assessment"
3. Line 75: "Besides, we should not erase the important "evidence" on the doctor's side by administering steroid eye drops or antibiotics before the diagnosis." I assume this sentence implies that pre-examination antibiotcs/steroid may affect the results of PCR. I think this sentence can be rephrased for better clarity.
4. Line 96: "the results always cannot be correct" should be deleted.
5. Line 101: "PCR as well as culture, information on the amount of pathogens is helpful in making proper decisions, e.g., when the results of high copy numbers of virus genomic DNA in the sample, it is possible that the virus should be pathogen for the disease." The whole sentence should be rephrased into "Regarding PCR or culture, information on the amount of pathogens (e.g., high copy numbers of a certain virus genomic DNA) provide important information in reaching the correct diagnosis.
6. Line 109: "...but the aqueous humor, rather than the corneal tissue, a doctor often submits for PCR testing." should be rephrased as "Therefore, PCR testing of the aqueous humor would be unrevealing, since cellular infiltration..."
7. Line 111: "Such sample collection may cause false negatives of the test, so caution is required, and doctors are responsible for the selection of samples to be collected." I think the whole sentence is redundant and can be deleted, since this issue has been clearly addressed in the previous sentences.
8. Line 123: "in the patient" should be deleted
9. Line 123: "Toxoplasma gondii that is the parasite" should be rephrased into "Toxoplasma gondii, on the contrary, ..."
10. Line 136: "We recommend that you validate your facilities and PCR machines, educate your staff and conduct inspections appropriately." For more formal wording, I suggest rephrase into "Validation of PCR machines in each institute, and education of the staff are mandatory"
11. Line 137: "We are doing everything possible to prevent bacterial/fungal contamination, such as devising the flow line in the inspection room, applying UV light to the work room and cabinets, and frequently changing gloves." rephrase into "For example, devising the flow line in the inspection room, applying UV light to the work room and cabinets, and frequently changing gloves are all practical measures to prevent the contaminations."
12. Line 164: delete the repeated "rRNA"
13. Line 166: What does "25F primers" mean?
Author Response
Comment and suggestion from Reviewer 1
1.Line 119: "As for Bartonella, even in cases of typical cat-scratch disease, the authors have no experience of PCR-positive intraocular fluids, and we are entrusted with the Bartonella serum antibody test." Beside their clinical experience, I think the authors should address a little more regarding previous evidence of the pathophysiology of CSD. Why is PCR mostly negative from vitreous/aqueous sampling ? (such as autoimmune reaction; molecular mimicry; or the pathogen is hidden deep in the retinal tissue ?)
Response: Thank you for your comments. Therefore, we revised the sentences as per your suggestions (revised manuscript, Lines 126-129).
2. Line 198: "Direct Strip PCR that skips DNA extraction" I suggest the authors cover a little more details on why DNA extraction can be skipped using this technique. Meanwhile, the reasons, pros and cons of not detecting bacteria and fungi by direct strip PCR should also be mentioned.
Response: We agree with your comments. Therefore, we added the sentences (revised manuscript, Lines 226-232).
Minor:
1.Abstract: "Infectious uveitis is a vision-threatening emergency..." As some infections are slowly progressive and not necessarily fulminant, I suggest using a more neutral description: "vision-threatening condition"
Response: We changed it as above sentence: "vision-threatening condition" (Line 12).
2. Line 60: "to perform multiplex PCR and graphically assess" I suggest "to perform multiplex PCR with graphical assessment"
Response: We have exchanged from "to perform multiplex PCR and graphically assess" to "to perform multiplex PCR with graphical assessment"(Line 69).
3. Line 75: "Besides, we should not erase the important "evidence" on the doctor's side by administering steroid eye drops or antibiotics before the diagnosis." I assume this sentence implies that pre-examination antibiotcs/steroid may affect the results of PCR. I think this sentence can be rephrased for better clarity. Response: As per your comments, we revised the sentences: It should be careful that pre-treatment specimens (e.g., by antibiotics/steroids) may affect PCR results (Lines 84-86).
4. Line 96: "the results always cannot be correct" should be deleted.
Response: As per your comments, we deleted that.
5. Line 101: "PCR as well as culture, information on the amount of pathogens is helpful in making proper decisions, e.g., when the results of high copy numbers of virus genomic DNA in the sample, it is possible that the virus should be pathogen for the disease." The whole sentence should be rephrased into "Regarding PCR or culture, information on the amount of pathogens (e.g., high copy numbers of a certain virus genomic DNA) provide important information in reaching the correct diagnosis.
Response: As per your comments, we have rephrased the sentences (Lines 109-111).
6. Line 109: "...but the aqueous humor, rather than the corneal tissue, a doctor often submits for PCR testing." should be rephrased as "Therefore, PCR testing of the aqueous humor would be unrevealing, since cellular infiltration..."
Response: We changed it as your sentences (Lines 116-117).
7. Line 111: "Such sample collection may cause false negatives of the test, so caution is required, and doctors are responsible for the selection of samples to be collected." I think the whole sentence is redundant and can be deleted, since this issue has been clearly addressed in the previous sentences.
Response: As per your comments, we deleted the sentences.
8. Line 123: "in the patient" should be deleted
Response: We deleted the sentences.
2
9. Line 123: "Toxoplasma gondii that is the parasite" should be rephrased into "Toxoplasma gondii, on the contrary, ..."
Response: Another reviewer (reviewer #3) described similar comments. So, we changed the sentences (Lines 13-131).
10. Line 136: "We recommend that you validate your facilities and PCR machines, educate your staff and conduct inspections appropriately." For more formal wording, I suggest rephrase into "Validation of PCR machines in each institute, and education of the staff are mandatory"
Response: We have rephrased the sentences (Line 144).
11. Line 137: "We are doing everything possible to prevent bacterial/fungal contamination, such as devising the flow line in the inspection room, applying UV light to the work room and cabinets, and frequently changing gloves." rephrase into "For example, devising the flow line in the inspection room, applying UV light to the work room and cabinets, and frequently changing gloves are all practical measures to prevent the contaminations."
Response: We have rephrased the sentences (Lines 145-147).
12. Line 164: delete the repeated "rRNA"
Response: We deleted the word.
13. Line 166: What does "25F primers" mean? Less...
Response: 25F Primer is just a primer name, so we deleted it from the text.
Thank you so much for helpful comments.

Reviewer 2 Report
Overview
A variety of pathogens can cause endophthalmitis, uveitis, and retinitis. Bacteria, fungi, parasites, and viruses are all potential pathogens. When inflammation is high, intraocular visualization becomes difficult and clinical signs cannot be observed; PCR can help confirm the diagnosis and allow for accurate and effective treatment. This review is timely and can be instructive to many readers.
<Introduction>
This paper outlines the use of PCR for diagnostics in the field of ophthalmology, and should touch on the history of how PCR has revealed the etiology of various ophthalmic diseases for which the cause was unknown. The authors' recent publication of citation #18, Adenovirus-Associated Uveitis with Necrotizing Retinitis, is shocking. Similarly, there are two discoveries in the field of ophthalmology that have been shocking in the past. The first was that HSV-2 was one of the pathogens causing retinitis; HSV includes HSV-1 and HSV-2, and HSV-2 was thought to cause sexually transmitted diseases. The other was that CMV was one of the pathogens causing corneal endotheliitis. Both of these were in the days before the development of modern, state-of-the-art equipment. Thanks to these discoveries, which have proven the utility of PCR, new methods and devices have continued to be developed. When the subject of PCR to identify the cause of ocular disease is discussed, the following two articles should be cited.
1. “High Prevalence of Herpes Simplex Virus Type 2 in Acute Retinal Necrosis Syndrome Associated with Herpes Simplex Virus in Japan.”
N. Itoh, N. Matsumura, A. Ogi, T. Nishide, Y. Imai, H. Kanai, et al.
Am J Ophthalmol 2000 129: 404-405
2. “Cytomegalovirus in aqueous humor from an eye with corneal endotheliitis”
Noriko Koizumi, Kenta Yamasaki, Satoshi Kawasaki, Chie Sotozono, Tsutomu Inatomi, Chikako Mochida, Shigeru Kinoshita
Am J Ophthalmol 2006 141: 564-5.
<2.2 interpretation and precautions of PCR results of eye specimen test>
Many viruses of the genus Herpesvirus are pathogens of retinitis. Herpesviruses of the genus are latent infections in the human body. When bleeding is seen, the intraocular vascular barrier breaks down and cells that have been circulating in the body flow into the eye. If those cells are latently infected with herpesviruses, they are detected by PCR, although they are not pathogens. Mention the possibility of contaminations that are not mere accidents in ophthalmic diseases caused by the genus herpesvirus. If there is a way to avoid it, please explain how. In PCR where quantification is possible, please indicate where to draw the line between positive and negative results.
<Table 1.>
Add items and subdivide items to make a better table. Add a line to indicate references cited. The pathogen section should be subdivided into bacteria, fungi, parasites, and viruses.
<Recent molecular biological methods for identifying infectious disease>
To make the review even better, it must cover the latest trends. nCounter® Analysis System (NanoString Technologies) should also be introduced. The following paper is suitable as a reference.
“An All-in-One Highly Multiplexed Diagnostic Assay for Rapid, Sensitive, and Comprehensive Detection of Intraocular Pathogens.”
P. J. M. Bispo, N. Belanger, A. Li, R. Liu, G. Susarla, W. Chan, et al.
Am J Ophthalmol 2023 Vol. 250 Pages 82-94
<Conclusions and future direction>
Koch's postulates should not be forgotten in infectious disease research. In the pursuit of speed and simplicity, PCR-based pathogen detection methods have advanced. Isolation and culture efforts should not be spared in order to include various pathogens in textbooks as etiologic agents of specific diseases. Currently and in the future. For the advancement of infectious disease science in the field of ophthalmology, it is desirable to mention this as well.
<references>
The year of publication of the cited reference #18, “Adenovirus-Associated Uveitis with Necrotizing Retinitis”, for which the author is the first author, is 2023. Errors in the bibliographies of the cited references will not be appreciated by this journal. Please double-check all other references.
Author Response
1
Comment and suggestion from Reviewer 2
<Introduction> This paper outlines the use of PCR for diagnostics in the field of ophthalmology, and should touch on the history of how PCR has revealed the etiology of various ophthalmic diseases for which the cause was unknown. The authors' recent publication of citation #18, Adenovirus-Associated Uveitis with Necrotizing Retinitis, is shocking. Similarly, there are two discoveries in the field of ophthalmology that have been shocking in the past. The first was that HSV-2 was one of the pathogens causing retinitis; HSV includes HSV-1 and HSV- 2, and HSV-2 was thought to cause sexually transmitted diseases. The other was that CMV was one of the pathogens causing corneal endotheliitis. Both of these were in the days before the development of modern, state-of-the-art equipment. Thanks to these discoveries, which have proven the utility of PCR, new methods and devices have continued to be developed. When the subject of PCR to identify the cause of ocular disease is discussed, the following two articles should be cited. 1. “High Prevalence of Herpes Simplex Virus Type 2 in Acute Retinal Necrosis Syndrome Associated with Herpes Simplex Virus in Japan.” N. Itoh, N. Matsumura, A. Ogi, T. Nishide, Y. Imai, H. Kanai, et al. Am J Ophthalmol 2000 129: 404-405 2. “Cytomegalovirus in aqueous humor from an eye with corneal endotheliitis” Noriko Koizumi, Kenta Yamasaki, Satoshi Kawasaki, Chie Sotozono, Tsutomu Inatomi, Chikako Mochida, Shigeru Kinoshita. Am J Ophthalmol 2006 141: 564-5.
Response: Thank you for your comments. We really appreciate. Therefore, in the introduction section, we have added similar sentences with new references (revised manuscript, Lines 37-49).
<2.3 interpretation and precautions of PCR results of eye specimen test>
Many viruses of the genus Herpesvirus are pathogens of retinitis. Herpesviruses of the genus are latent infections in the human body. When bleeding is seen, the intraocular vascular barrier breaks down and cells that have been circulating in the body flow into the eye. If those cells are latently infected with herpesviruses, they are detected by PCR, although they are not pathogens. Mention the possibility of contaminations that are not mere accidents in ophthalmic diseases caused by the genus herpesvirus. If there is a way to avoid it, please explain how. In PCR where quantification is possible, please indicate where to draw the line between positive and negative results.
Response: As your comments, even with quantitative PCR, there are often cases where it is difficult to draw the line between positive and negative results. Therefore, we added the following sentences to the revised manuscript: In addition to issues of false-positive or false-negative cases by PCR, we must think about the line between positive and negative results. If ocular tissues and infiltrating cells are latently infected with a virus (e.g., herpesviruses), they are detected by PCR, although they are not pathogens. When quantitative PCR is performing, that is "positive" when high copy number's DNA is present in the sample. However, in the case of low copy numbers, we are unable to distinguish between positive and negative results in order to diagnose the infectious pathogens. However, this is not only a problem with PCR tests, but also with other tests (e.g. virus antigen tests). Rather than using the results of PCR alone for diagnosis, we believe that characteristic ocular findings with active inflammation, other test results, and other information such as treatment effects should be comprehensively evaluated (Lines 153-163)
<Table 1.> Add items and subdivide items to make a better table. Add a line to indicate references cited.
The pathogen section should be subdivided into bacteria, fungi, parasites, and viruses. <Recent molecular biological methods for identifying infectious disease>
To make the review even better, it must cover the latest trends. nCounter® Analysis System (NanoString Technologies) should also be introduced. The following paper is suitable as a reference. “An All-in-One Highly Multiplexed Diagnostic Assay for Rapid, Sensitive, and Comprehensive Detection of Intraocular Pathogens.”
P. J. M. Bispo, N. Belanger, A. Li, R. Liu, G. Susarla, W. Chan, et al. Am J Ophthalmol 2023 Vol. 250 Pages 82-94
2
Response: As per your suggestion, we have inserted the content of the reference paper for the nCounter® analysis system (NanoString Technologies) into new Table 1.
<Conclusions and future direction>
Koch's postulates should not be forgotten in infectious disease research. In the pursuit of speed and simplicity, PCR-based pathogen detection methods have advanced. Isolation and culture efforts should not be spared in order to include various pathogens in textbooks as etiologic agents of specific diseases in currently and in the future. For the advancement of infectious disease science in the field of ophthalmology, it is desirable to mention this as well.
Response: Actually, Koch's postulates is my favorite, and I think about it all the time. We therefore inserted this similar sentence in <Conclusions and future direction >: For the advancement of infectious disease science in the field of ophthalmology, Koch's postulates should not be forgotten in infectious disease research. In the pursuit of speed and simplicity, PCR-based pathogen detection methods have advanced. Isolation and culture efforts should not be spared in order to include various pathogens in textbooks as etiologic agents of specific diseases in the future. If possible, animal experiments are also necessary. Basic research on infectious diseases is extremely important for further development of clinical testing (Lines 413-419).
<references>
The year of publication of the cited reference #18, “Adenovirus-Associated Uveitis with Necrotizing Retinitis”, for which the author is the first author, is 2023. Errors in the bibliographies of the cited references will not be appreciated by this journal. Please double- check all other references.
Response: We checked the references through the revised manuscript.

Reviewer 3 Report
General comments
1. The authors have submitted a very interesting and comprehensive review of the recent role of PCR testing to diagnose infectious ocular diseases. It is likely that English is not the first language of the authors and the manuscript will require language editing.
2. The tone of the manuscript is rather conversational in places and should please be checked when English language editing is performed.
3. This reviewer was unable to view the Supplementary movies.
4. Line 134 and elsewhere: Propionibacterium acnes is now called Cutibacterium acnes. Please correct throughout.
Specific comments:
1. The title should be changed to "Role of recent PCR tests for infectious ocular diseases: From laboratory-based studies to the clinic"
2. Line 26: Please consider replacing the word "wiping" with "swabbing".
3. Line 35: please consider deleting "the most".
4. Line 44: please consider replacing the word "eye" with "ocular"
5. Lines 47 - 54: please make the legend more succinct and consider omitting the sentence about the disposable pipette.
6. Line 81: please add the word "clinic" after "outpatient".
7. Lines 119 - 121: please consider omitting the sentence about Bartonella as it does not add much value.
8. Line 123: The sentence starting with Toxoplasma gondii is confusing as it seems to suggest the toxocariasis is caused by T. gondii. Please clarify.
9. Lines 136 - 139: Please refrain from using the first person in this type of manuscript.
10. Line 143: please consider replacing "indigenous" with "commensal".
11. Line 145: please consider replacing "resident bacterium" with "commensal" or "commensal organism".
12. Line 148: please avoid the first person here and throughout the text
13. Line 157: please add M. before tuberculosis
14. Line 162: please add the words "exist" after "techniques" to clarify the sentence
15. Line 165: please write out the abbreviation "bp" at first use.
16. Line 168: please state what "BLAST" stands for.
17. Lines 223 - 226: please make the legend shorter and more concise.
18. Lines 232 - 235: please check the grammar and intended meaning.
19. Line 276: please change "symptoms" to "signs".
Author Response
1
Comment and suggestion from Reviewer 3
1. The authors have submitted a very interesting and comprehensive review of the recent role of PCR testing to diagnose infectious ocular diseases. It is likely that English is not the first language of the authors and the manuscript will require language editing.
Response: Thank you for your comments. After all of the reviews, we will take the English editing by a native speaker.
2. The tone of the manuscript is rather conversational in places and should please be checked when English language editing is performed.
Response: Thank you for the comments.
3. This reviewer was unable to view the Supplementary movies.
Response: Other reviewers and editors were able to see it, so we will see how it goes.
4. Line 134 and elsewhere: Propionibacterium acnes is now called Cutibacterium acnes. Please correct throughout.
Response: We have rephrased to Cutibacterium acnes or C. acnes through the manuscript.
Specific comments:
1. The title should be changed to "Role of recent PCR tests for infectious ocular diseases: From laboratory-based studies to the clinic"
Response: We agree with your comments. So, we changed the title.
2. Line 26: Please consider replacing the word "wiping" with "swabbing".
Response: We have replaced the word (Line 25).
3. Line 35: please consider deleting "the most".
Response: We have deleted the words.
4. Line 44: please consider replacing the word "eye" with "ocular"
Response: We have replaced the word.
5. Lines 47 - 54: please make the legend more succinct and consider omitting the sentence about the disposable pipette.
Response: We have replaced the sentences in the legend.
6. Line 81: please add the word "clinic" after "outpatient".
Response: We have added the word (Line 90).
7. Lines 119 - 121: please consider omitting the sentence about Bartonella as it does not add much value.
Response: We were instructed by another reviewer to add a thoughtful explanation to this sentence, so we left it in with additional sentences.
8. Line 123: The sentence starting with Toxoplasma gondii is confusing as it seems to suggest the toxocariasis is caused by T. gondii. Please clarify.
Response: We have replaced the sentences (Lines 130-133).
9. Lines 136 - 139: Please refrain from using the first person in this type of manuscript.
Response: Another reviewer made a similar observation and changed the sentences in the text.
10. Line 143: please consider replacing "indigenous" with "commensal".
Response: We have replaced the word (Line 150).
11. Line 145: please consider replacing "resident bacterium" with "commensal" or "commensal organism".
Response: We have replaced into "commensal organism" (Line 152).
12. Line 148: please avoid the first person here and throughout the text
Response: We have replaced the sentences to avoid the first person.
13. Line 157: please add M. before tuberculosis
Response: We have added the word, “M” (Line 176).
14. Line 162: please add the words "exist" after "techniques" to clarify the sentence
Response: We have added the word, "exist" (Line 181).
15. Line 165: please write out the abbreviation "bp" at first use.
Response: We have written out “bp” to “base-pair” (Line 184).
16. Line 168: please state what "BLAST" stands for. Response: We have added the sentences, Basic Local Alignment Search Tool (BLAST) analysis in order to explain it (Lines 187-192).
2
17. Lines 223 - 226: please make the legend shorter and more concise.
Response: We have replaced the legend shorter.
18. Lines 232 - 235: please check the grammar and intended meaning.
Response: We have changed the sentences.
19. Line 276: please change "symptoms" to "signs".
Response: Figure 3 has been removed and replaced with text only. So, we also omitted the legend including "symptoms". Thank you for your helpful comments.

Round 2
Reviewer 1 Report
The authors have made a thorough revision. I have only one comment:
Line 84: It should be careful that pre-treatment specimens (e.g., by antibiotics/steroids) may affect PCR results.
Please rephrase into --> It should be "kept in mind" thatAuthor Response
Line 84: It should be careful that pre-treatment specimens (e.g., by antibiotics/steroids)
may affect PCR results.
Please rephrase into --> It should be "kept in mind" that
Response: Thank you for your comments. Therefore, we revised the sentences as
per your suggestions (revised manuscript, Lines 84-85).
